# Dipsticks and point-of-care Microscopy to reduce antibiotic use in women with an uncomplicated Urinary Tract Infection (MicUTI): protocol of a randomised controlled pilot trial in primary care

Peter K Kurotschka [1], Gábor Borgulya [1], Eva Bucher,[1] Isabell Endrich,[1] Adolfo Figueiras,[2] Jochen Gensichen,[3] Alastair D Hay,[4] Alexander Hapfelmeier,[5,6] Christian Kretzschmann,[1] Oliver Kurzai,[7] Thien-Tri Lam,[7] Orietta Massidda,[8] Linda Sanftenberg [3], Guido Schmiemann,[9] Antonius Schneider [5], Anne Simmenroth,[1] Stefanie Stark,[10] Lisette Warkentin [10], Mark H Ebell [11], Ildikó Gágyor [1], on behalf of the Bavarian Practice-Based Research Network (BayFoNet)

For numbered affiliations see end of article.

**Correspondence to**
Dr Peter K Kurotschka;
kurotschka_p@ukw.de

## ABSTRACT

**Introduction** Uncomplicated urinary tract infections (uUTIs) in women are common infections encountered in primary care. Evidence suggests that rapid point-of-care tests (POCTs) to detect bacteria and erythrocytes in urine at presentation may help primary care clinicians to identify women with uUTIs in whom antibiotics can be withheld without influencing clinical outcomes. This pilot study aims to provide preliminary evidence on whether a POCT informed management of uUTI in women can safely reduce antibiotic use.

**Methods and analysis** This is an open-label two-arm parallel cluster-randomised controlled pilot trial. 20 general practices affiliated with the Bavarian Practice-Based Research Network (BayFoNet) in Germany were randomly assigned to deliver patient management based on POCTs or to provide usual care. POCTs consist of phase-contrast microscopy to detect bacteria and urinary dipsticks to detect erythrocytes in urine samples. In both arms, urine samples will be obtained at presentation for POCTs (intervention arm only) and microbiological analysis. Women will be followed-up for 28 days from enrolment using self-reported symptom diaries, telephone follow-up and a review of the electronic medical record. Primary outcomes are feasibility of patient enrolment and retention rates per site, which will be summarised by means and SDs, with corresponding confidence and prediction intervals. Secondary outcomes include antibiotic use for UTI at day 28, time to symptom resolution, symptom burden, number of recurrent and upper UTIs and re-consultations and diagnostic accuracy of POCTs versus urine culture as the reference standard. These outcomes will be explored at cluster-levels and individual-levels using descriptive statistics, two-sample hypothesis tests and mixed effects models or generalised estimation equations.

**Ethics and dissemination** The University of Würzburg institutional review board approved MicUTI on 16

## STRENGTHS AND LIMITATIONS OF THIS STUDY

⇒ This study will inform the design and conduct of future confirmatory randomised controlled trials (RCTs) to evaluate the effects of a point-of-care test-guided management strategy on the use of antibiotics in women with suspected uncomplicated urinary tract infection in primary care settings.

⇒ The open-label cluster-RCT design implies that participants cannot be identified before randomisation, thus, post-randomisation recruitment may represent a source of selection bias.

⇒ The execution and interpretation of microscopy test results may potentially differ across medical assistants, affecting diagnostic accuracy measurements and treatment outcomes. This will be counteracted by providing all intervention practices with standardised ad hoc competency-based training.

December 2022 (protocol n. 109/22-sc). Study findings will be disseminated through peer-reviewed publications, conferences, reports addressed to clinicians and the local citizen's forums.

**Trial registration number** ClinicalTrials.gov NCT05667207.

## INTRODUCTION

Uncomplicated urinary tract infections (UTIs) are a common reason for adult women to consult a healthcare professional and a major driver of antibiotic prescriptions in the community.[1] Up to 95% of women presenting to their general practitioner (GP) receive antibiotics, despite

uncomplicated UTIs (uUTIs) being self-limiting in up to 50% of cases.[1 2]

As antibiotic prescriptions are closely related to the emergence of resistant micro-organisms, reducing unnecessary use of these drugs in uUTI management is important to counteract the rise of antimicrobial resistance.[3–5]

Randomised controlled trials (RCTs) have been conducted to establish whether alternatives to immediate antibiotic treatment, such as delayed prescriptions, herbal formulations or non-steroidal anti-inflammatory drugs reduce antibiotic use without significant patient harm.[6–11] All strategies were highly effective in reducing antibiotic use over usual care. Nonetheless, if compared with immediate antibiotics, the alternative treatment strategies showed an increased risk of longer symptom duration, higher symptom burden, incomplete recovery, febrile UTI, pyelonephritis and antibiotic use at follow-up.[12]

A recent meta-analysis of RCTs confirmed these findings analysing individual participant data of 3524 patients from eight primary care trials.[13] In addition, the authors focused on prognostic and moderating factors of treatment effects. They found that, in women treated without antibiotics, incomplete recovery was more likely if, at baseline, erythrocytes were found in urine and women had a positive urine culture (OR 4.68, Bayesian credible interval (CI) 2.07 to 10.77). If only the urine culture was positive, an incomplete recovery was likely with OR 2.24 (CI 1.00 to 5.17), while if only the test for erythrocytes was positive the OR was 2.60 (CI 1.086 to 32). In contrast, the authors found no significant difference in the likelihood of incomplete recovery between non-antibiotic strategies and immediate antibiotics if both tests were negative (OR 0.82, CI 0.331 to 95). Moreover, positive urine culture results and erythrocytes in urine were found to be independent predictors of complications such as pyelonephritis, febrile UTI and of antibiotic use at follow-up.[13]

These findings show a clear benefit of immediate antibiotics in all cases in which the woman's urine culture is positive, and erythrocytes are found in urine. Simple and affordable tests exist to detect erythrocytes at the point-of-care (POC) rapidly, namely urinary dipsticks[14]; however standard urine cultures are not useful for immediate clinical decisions, as they require 48 hours or more to provide definitive results.[15–17] This led to studies evaluating portable devices to perform urine culture, which could deliver quicker results than standard cultures and could possibly lead to a more accurate diagnosis than the usual diagnostic approach based only on symptoms and dipstick test results. The multinational POETIC RCT found that POC urine culture and susceptibility testing were not effective at reducing immediate antibiotic prescriptions.[18] A Danish RCT found that adding POC susceptibility testing to POC-culture was unlikely to improve the appropriateness of treatment decisions in women with uUTI.[19] One explanation for these poor results may be, that POC-cultures require 24 hours to deliver results and, given the prognostic and moderating effects of a positive urine culture

(ie, significant bacteriuria) the treatment decision is best to be made immediately during index consultation.[13]

To date, urine microscopy is the only rapid POC-test (POCT) able to directly detect bacteria in urine that has been evaluated in a general practice setting. Among the various microscopic methods, phase-contrast microscopy does not require centrifugation or other time-consuming specimen preparation methods, thus being easier to implement in practice than other microscopic approaches.[20] Several studies have investigated its diagnostic accuracy in detecting UTI in adult outpatients and have shown promising, although conflicting results. Reported sensitivity varied between 74% and 95%, while specificity ranged between 63% and 97%.[21–24] Nevertheless, although a recent systematic review concluded that microscopy 'seems to be an accurate, valid and feasible screening-test for bacteriuria in patients with symptoms of UTI in general practice', it acknowledged that there is still a lack of solid evidence supporting its accuracy to diagnose UTIs.[25]

Diagnostic RCTs have the advantage over traditional diagnostic studies of providing information on test accuracy but also on whether the expected differences in accuracy over usual care are clinically relevant.[26] To date, the effects of a diagnostic and treatment strategy guided by POCTs for bacteria and erythrocytes in urine on antibiotic use in women with suspected uUTI have not been studied. This pilot RCT aims to provide information about the feasibility and sample size of a full-scaled diagnostic RCT in general practice and to provide initial data on the effects of a POCT–guided diagnostic and treatment algorithm on antibiotic use. POCT to be used are phase-contrast microscopy to detect bacteria and urinary dipsticks to detect erythrocytes in urines.

## METHODS AND ANALYSIS
### Design
MicUTI (Microscopy in Urinary Tract Infections) is a pragmatic open-label, two-arm parallel cluster randomised pilot trial. General practices are randomly allocated to either the POCT-based management strategy (intervention) or standard care management. GPs whose practice is allocated to the intervention will have their management guided by phase-contrast microscopy to identify bacteria and urinary dipsticks to identify erythrocytes in urine samples. GPs whose practice is allocated to the control arm will perform usual care, with treatment decisions-based mainly on symptoms and dipstick testing for erythrocytes, leucocytes and nitrites. For the trial duration (6 months), practices are asked to systematically recruit women aged 18–70 years with symptoms suggesting a uUTI (dysuria, frequency/urgency of micturition, nocturia, with or without lower abdominal pain) to be included in the trial.

### Objectives
#### Primary objective
The primary objective of this study is to assess the feasibility of a full-scaled diagnostic RCT. The primary

endpoints are patient recruitment efficacy and the percentage of retention in the trial. Recruitment efficacy will be calculated as the number of participants enrolled per site over the 6 months of trial duration, and retention as the percentage of complete follow-ups per site over 28 days. A confirmatory study without major design changes will be considered feasible if study sites recruit 10 patients each and patients lost to follow-up do not exceed 20%.

### Secondary objectives

The secondary objectives of the MicUTI trial are to compare the intervention and control practices with regard to the following endpoints:

1. Total antibiotic use: number of antibiotic prescriptions per patient with UTI within 28 days.
2. Defined daily doses of the prescribed antibiotics per patient with UTI within 28 days.
3. Inappropriate antibiotic use: the percentage of patients with symptoms of UTI who were prescribed antibiotics among those with negative urine culture findings.
4. Number of immediate and delayed antibiotic prescriptions for uUTI at the initial consultation.
5. Number of early relapses of UTI (days 0–14).
6. Number of recurrent UTIs (days 15–28).
7. Number of upper UTIs within 28 days.
8. Number of consultations due to UTI (or symptoms of UTI) within 28 days.
9. Time to complete symptom resolution is defined as a maximum of 1 point in each of the Urinary Tract Infection-Symptom and Impairment Questionnaire (UTI-SIQ-8) items.[27]
10. Total symptom burden on days 0–7 (area under the curve of the UTI-SIQ-8 total symptom score).
11. Diagnostic accuracy of microscopy with or without concomitant dipstick test compared with the reference standard test (urine culture).

A further objective of this study is to evaluate the feasibility and acceptability of POC-microscopy to detect bacteria in urine and the training sessions addressed to medical assistants in the intervention arm practices.

### Setting

The MicUTI study is conducted in a primary care research network based in the German Federal State of Bavaria, the Bavarian Practice-Based Research Network (BayFoNet). Three out of the five university departments part of the BayFoNet participate in MicUTI. These are the Department of General Practice of the University Hospital Würzburg (hereafter abbreviated as UKW), which is the trial coordinating centre, the Institute of General Practice of the Friedrich-Alexander Universität Erlangen-Nürnberg (hereafter abbreviated as FAU) and the Institute of General Practice and Family Medicine of the Ludwig-Maximilians University Hospital Munich (hereafter abbreviated as LMU), who collaborate in the recruitment of study sites and data collection. An accompanying mixed-methods process evaluation assesses the implementation of this trial into the BayFoNet and further investigates its feasibility and acceptability to general practice teams. Details of the process evaluation study are published elsewhere.[28]

### Recruitment of study sites

The UKW and the FAU will recruit 20 general practices located in the Federal State of Bavaria (Würzburg and Erlangen and surrounding areas) to participate in the trial through postal invitations, institutional advertisements such as newsletters and website announcements and public conferences with GP practice teams.

### Randomisation of study sites

General practices participating in the trial (hereafter: study sites) will be asked to provide the number of consulting patients over the last year to determine practice size. Study sites of larger size are foreseeably able to recruit more patients for the duration of the trial, so that practice size will be balanced between the two arms (each with 10 practices) using minimisation.[29] The minimisation algorithm is centrally programmed in Python by the study statistician. To mask the trial statistician (GB) to intervention/control assignments, the minimisation algorithm will be executed by the trial's statistical consultant based at the Technical University of Munich (AH), who is not involved in the conduction of the trial. Intervention/control assignments of practices will be communicated to the UKW and the FAU, who then will communicate their allocation to the study sites. To minimise selection bias, study sites were identified for inclusion in the trial prior to randomisation.

### Patient recruitment/enrolment
#### Identification of women eligible for inclusion

At presentation, at all study sites, medical assistants will consecutively approach all adult women presenting with two or more symptoms suggestive of acute uUTI (dysuria, frequency, urgency, nocturia, lower abdominal pain) to identify trial participants. To ensure consecutive enrolment and control for selection bias, all potential participants will be approached and listed in paper-based anonymous patient pre-screening logs at each study site. Interested women will be asked to complete the validated symptom questionnaire UTI-SIQ-8[27] and will be provided with a comprehensive patient information leaflet that details trial procedures in lay language.

The GP will exclude from participation all women with one or more of the following:

► Signs of a complicated UTI (anamnesis of fever, chills or flank pain).
► Clinically relevant immunosuppression (ie, current use of any immunosuppressive therapy, congenital or acquired disorders of immunity).
► Acute or chronic functional or anatomical variations in the urinary tract except for chronic kidney failure with a glomerular filtration rate (eGFR) >45 mL/min.

- ► Permanent bladder catheter or use of bladder catheter within the past 2 weeks.
- ► UTI within the past 2 weeks.
- ► Use of any antibiotic within the past 2 weeks.
- ► Accommodation in a nursing home or hospital stay within the past 2 weeks.
- ► Severe neurologic, psychiatric illness, severe dementia or severe substance use disorder.
- ► Other severe diseases.
- ► Being unable to understand the informed consent or to complete the patient diary.
- ► Known pregnancy.

Women who consent will be asked to sign an informed consent form, which also contains data protection regulations.

### Intervention arm

GPs whose practice is allocated to the intervention will have their management guided by POCTs, namely phase-contrast microscopy and urinary dipsticks, for all patients consenting to participation.

### Point-of-care tests

Medical assistants will perform microscopy using phase-contrast microscopes ('Primostar', Carl Zeiss Suzhou, Suzhou, China) to examine 7 µL of clean-catch midstream urine (MSU) without prior centrifugation at 400× magnification in a precision counting chamber (Fast-Read 102 slides, Biosigma S.r.l., Cona (VE), Italy) to detect bacteria. A test will be considered positive if more than a few bacteria of the same shape (rod or cocci) or if many bacteria of different shapes (rod and cocci) are detected per high power field.[30]

Urinary dipstick analysis will be performed using COMBUR5-Test (Roche Diagnostics GmbH, Mannheim, Germany) to examine 50 mL MSU to identify erythrocytes. In accordance with the manufacturer instructions, the COMBUR5-test strip will be dipped in the urine sample for about 1 s, wiped against the rim of the vessel to remove excess urine and read manually after 60 s by comparing the colour of the detection pad of the strip with the colour scale on the test strip vial. A positive test result is defined as the colour change of the strip corresponding to 1+erythrocytes or greater.

All the analyses will be conducted immediately after the MSU specimen collection at each study site.

### Training of study sites in trial procedures

Medical assistants will be provided with ad hoc training to approach potentially eligible women consecutively, and to perform POCTs according to the study protocol by dedicated staff. Training will be delivered face-to-face prior to patient recruitment over a 3-hour session (baseline session), as well as 3 months after the start of patient recruitment (refresher session), following the principles of competency-based medical education.[31] The baseline training session will entail the following: (1) Guideline knowledge on UTI in women to allow medical assistants to identify potentially eligible women in each study site; (2) POCT training to perform and interpret the results of phase-contrast microscopy and urinary dipsticks according to the study protocol. The knowledge and skills gained, and their retention, as well as the feasibility and the acceptability of the training sessions and the use of phase-contrast microscopes in daily practice, will be assessed through a questionnaire addressed to medical assistants before, immediately after, and 3 months after the training session and through face-to-face semi-structured interviews, 3 months after the training session. Interviews will be conducted following a pre-planned interview-guide informed by the Consolidated Framework for Implementation Research.[32] Practical tests at the microscope directly after the training session and after 3 months will assess each medical assistant's acquisition and retention of practical skills.

### Point-of-care test guided treatment algorithm

Based on the prognostic and moderating effects on treatment outcomes of erythrocytes and bacteria in urine,[13] GPs will be encouraged to apply the following treatment algorithm (figure 1) to consenting women, taking their preferences into account:

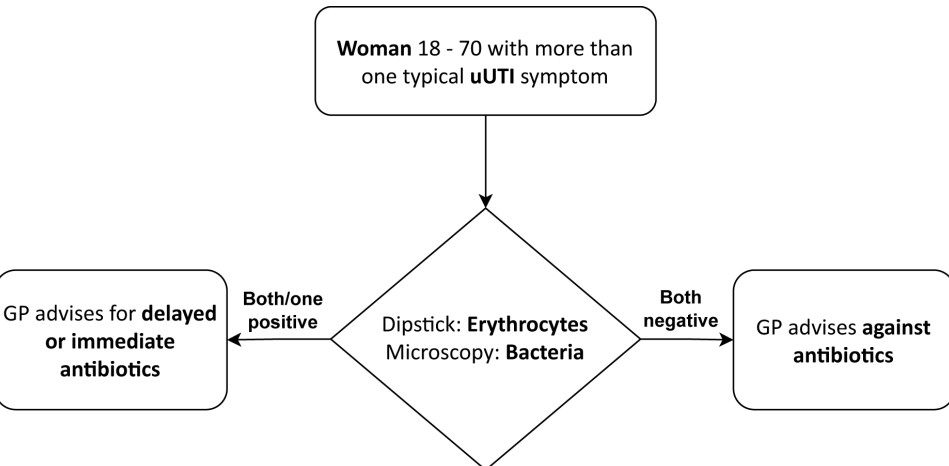

**Figure 1** Treatment algorithm in the intervention arm. GP, general practitioner; uUTI, uncomplicated urinary tract infection.

1. If POCTs are positive for bacteria by microscopy and/or for erythrocytes by dipsticks, the GP issues, at their own clinical judgement, a delayed or immediate prescription for an antibiotic. In the MicUTI intervention, a delayed prescription can be issued by GPs, at their discretion, in either way: the prescription is handed out to the patient with the advice to redeem it only if symptoms do not improve or worsen in 48 hours, or it is withheld in the GP practice. It can be handed out if the patient presents again with issues.

2. If POCTs are negative for bacteria and erythrocytes, the GP advises for self-help remedies according to national guidelines and to do without antibiotics.[33 34]

### Data collection
#### Baseline
Baseline data will be collected during routine consultation. The medical assistant records microscopy and dipstick test results, clinical features and the GP's management decisions, including antibiotics prescribed on a baseline paper-based case report form (CRF). In addition, the medical assistant inoculates a dipslide (see paragraph 'microbiological analyses') to be sent via post on the same day to the central research laboratory based at the Institute of Hygiene and Microbiology of the Julius-Maximilians-University of Würzburg (IHM) for microbiological analysis.

#### Follow-up
Follow-up data will be collected through a self-directed patient diary, follow-up telephone calls on day 28 from inclusion and follow-up in the GP practice in case of reconsultation. In addition, an electronic medical records (EMR) review will also be performed to double-check for missing follow-up data.

**Patient diary:** Women will be asked to rate their daily symptoms and impairment due to UTI on a scale from 1 (no symptoms/impairment at all) to 5 (very strong symptoms/impairment) and to write down the antibiotics taken in a diary for at least 7 days (or up to 14 days if symptoms last longer).[27] Enough space in the diary is left for women to take extra notes. They will be asked to return the completed diary to their GP practice using a pre-paid envelope or in person. Research nurses at the FAU and the LMU will perform telephone calls on days 2–4 from inclusion to remind enrolled women to fill out and return the diary.

**Follow-up telephone calls:** Calls will be performed by research nurses at the FAU and the LMU on day 28 from inclusion to collect data on symptoms, antibiotic use due to UTI, recurrences and reconsultations with UTI symptoms.

**Follow-up in the GP practice:** Every time a woman reconsults for UTI within 28 days, the GP records clinical features, management decisions and antibiotics prescribed on follow-up paper-based CRFs.

**Electronic medical records review:** Researchers from the UKW and the FAU will perform outreach visits at the end of the patient recruitment (6 months) in each of the study sites. Paper-based CRFs will be collected, and data will be controlled for completeness. To double-check data entries for accuracy and completeness, EMRs will be enquired on reconsultations with UTI symptoms, recurrences, antibiotics prescribed due to UTI and upper UTI until day 28 from inclusion.

### Control arm–usual care
Practices in the control arm will not necessarily have their management guided by POCTs. They will perform usual care. The treatment decision is usually based on symptoms. Dipsticks for detecting erythrocytes, leucocyte esterase or nitrites can be added in case of diagnostic uncertainty.[33] Procedures are the same as those outlined above, except for POCTs: microscopy only pertains to the intervention arm.

Table 1 and figure 2 summarise trial procedures.

### Microbiological analyses
For cultural urine diagnostics, at each study site, medical assistants immerse a dipslide, that is, a culture medium carrier (Uricult Plus, Roche Diagnostics GmbH, Mannheim, Germany) into the test urine until the agar surface is completely covered. After removal from the urine, the carrier is placed into a sterile transport tube for transportation via post to the IHM. On arrival, the carrier is incubated at $35\pm2°C$ for 16–24 hours. If bacterial growth is detected, the subsequent analysis is carried out according to the established diagnostic procedures of the accredited medical laboratory. Briefly, the analysis includes the determination of the overall bacterial count on the Cystine Lactose Electrolyte Deficient–agar plates, and those of gram-negative bacteria (on MacConkey agar plates) and of *Enterococcus* spp (on the *Enterococcus* agar plates). Bacteria are subcultured on Columbia Sheep Blood agar plates and MacConkey agar plates to be subsequently identified by mass spectrometry (VITEK MS, BioMérieux, Marcy-l'Etoile, France). Susceptibility testing of pathogenic bacteria is achieved using VITEK 2 (BioMérieux, Marcy-l'Etoile, France) and interpreted according to the European Committee on Antimicrobial Susceptibility Testing (EUCAST) criteria.[35]

To define a significant growth in the MSU specimens, the criteria specified in the updated 2020 German infectious disease and microbiology laboratory quality standards will be used.[30]

### Statistical analyses
Descriptive statistics will be calculated for all variables per site and as summary measures across sites. Continuous and count variables will be summarised by mean and SD, or median and IQR in case of skewed data. Binary and categorical variables will be summarised through counts and percentages. CIs and prediction intervals will be calculated for the two feasibility endpoints (recruitment efficacy and percentage of retention in the trial) in addition.

**Table 1** Schedule of Microscopy in Urinary Tract Infections trial procedures according to the Standard Protocol Items: Recommendations for Interventional Trials guideline[40]

| Time point (day 0=patient enrolment) procedures | <0 | Day 0 Baseline | Day 1–7 | Day 28 | End of patient enrolment* |
|---|---|---|---|---|---|
| Enrolment: | | | | | |
| Practice recruitment | X | | | | |
| Randomisation | X | | | | |
| Allocation | X | | | | |
| Training | X | | | | |
| Screening for potential inclusion | | X | | | |
| Formal eligibility assessment | | X | | | |
| Intervention: | | | | | |
| Baseline consultation† | | X | | | |
| POCTs‡ | | X | | | |
| Urine sent to laboratory† | | X | | | |
| Assessments: | | | | | |
| Practice size | X | | | | |
| Paper-based CRF§ | | X | | | |
| Patient diary¶ | | | X | | |
| Follow-up telephone calls | | | | X | |
| EMR review** | | | | | X |

*End of patient enrolment in each practice.
†Applies to intervention and control arm practices.
‡Applies only to intervention arm practices.
§In case of reconsultation for UTI on days 1–28, paper-based follow-up CRFs are used.
¶Follow-up on days 1–7 or up to day 14 if symptoms last longer or recur.
**Follow-up on days 1–28 (CRF completeness and accuracy check).
CRF, case report form; EMR, electronic medical records; POCT, point-of-care test; UTI, urinary tract infection.

We acknowledge that this feasibility RCT is not powered to detect any statistically significant effects. Nevertheless, explorative analyses will be undertaken at the cluster-level and the individual-level to evaluate the effects of the intervention on the secondary outcome variables using two-sample hypothesis tests, random effects models and generalised estimation equations, as appropriate. Alongside point-estimates, p values and 95% CIs will be reported.

Missing data will be explored to understand why they are unavailable; when missing data are due to incomplete follow-up survival or censored regression analyses will be considered, otherwise multiple imputation with chained equations or Bayesian imputation will be applied. The intraclass correlation coefficient will be calculated for the secondary endpoints to inform subsequent sample size and power calculations.

All the analyses will be undertaken using statistical software packages Stan,[36] R and Stata (StataCorp College Station, Texas, USA: StataCorp).

### Qualitative data analysis of interviews addressed to medical assistants

All interviews will be de-identified, transcribed verbatim and analysed by at least two researchers inductively via content-analysis inspired by Kuckartz.[37] The analysts perform first line-by-line coding after extensive reading of the interviews (familiarisation). While iteratively refining the code labels, they are grouped to generate categories and subcategories. In subsequent iterations of the process, categories and subcategories are refined in light of the research question and are summarised in a final report of the analyses.

### Sample size considerations

An ad hoc analysis undertaken on claims data provided by the Bavarian health insurance union (Kassenärztliche Vereinigung) covering all diagnosed UTIs in ambulatory care from 2015 to 2019 showed that each GP had an average of 60 encounters for uUTI in women aged 18–70 per year, with only slight seasonal fluctuations (data not shown). Considering that some of these encounters occur in out-of-hours care and that not all of the patients presenting with uUTI are going to be eligible, we expect that 200 patients in both arms (ie, 100 patients per arm=an average of 10 enrolled patients in each of the 20 clusters) is a reasonably achievable sample within 6 months of trial duration.

Assuming that the percentage of complete follow-up is 75% in the population the patients are sampled from,

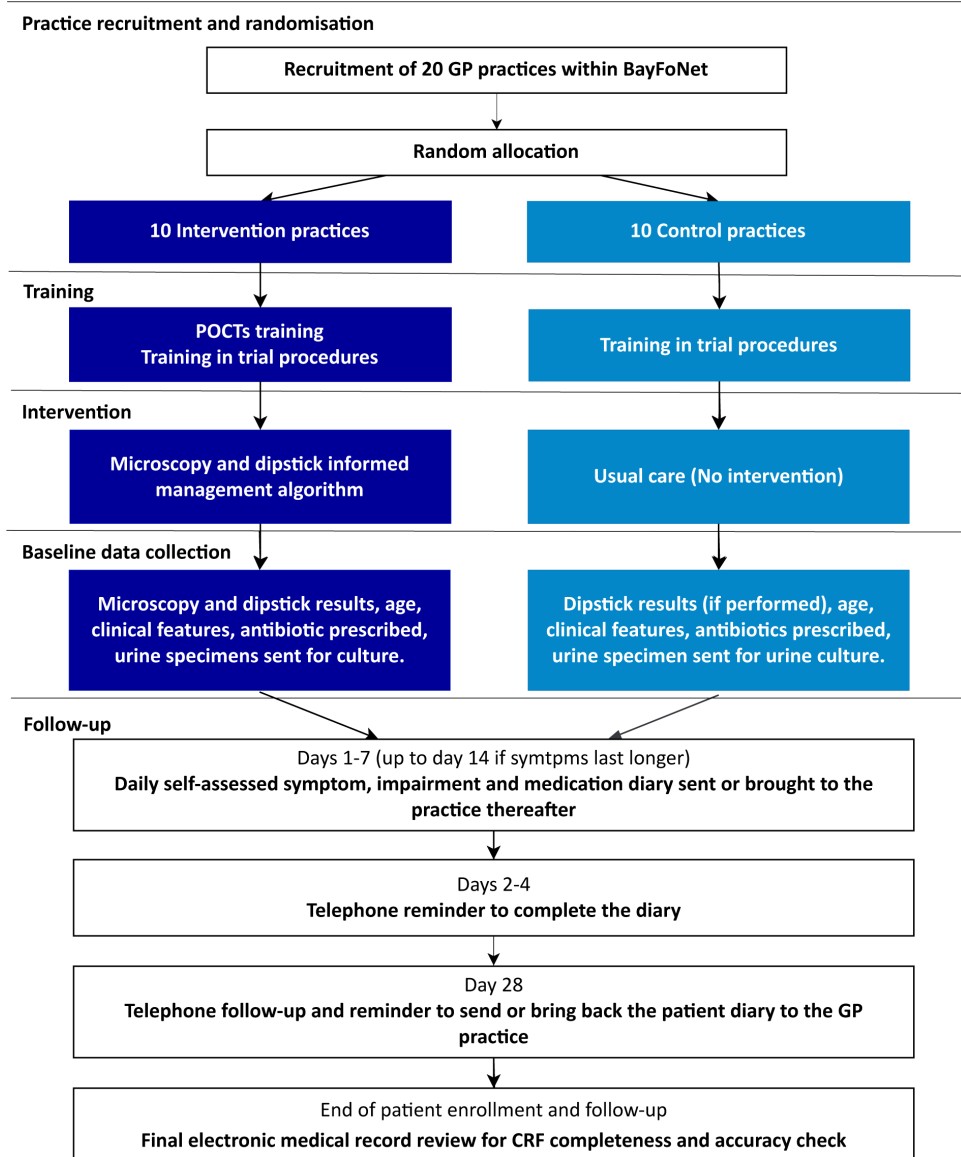

**Figure 2** Flow diagram of the Microscopy in Urinary Tract Infections trial procedures. BayFoNet, Bavarian Practice-Based Research Network; CRF, case report form; GP, general practitioner; POCTs, point-of-care tests.

a simulation study showed that the 200 patients of the feasibility trial enable the estimation of the percentage of complete follow-up with a 95% CI of width of less than 13% with a likelihood of 0.90.

## Patient and public involvement
A citizens' forum was established within the BayFoNet framework. Invited are all citizens who are interested in contributing to research projects of the Department of General Practice (University Hospital Würzburg). All important steps that are necessary for different projects and studies are discussed with the participants.

The MicUTI study has been discussed with the citizens twice so far. After the presentation of the study, the participants discussed whether therapy for uUTI without antibiotics would be conceivable. In addition, the members gave feedback on the comprehensibility and clarity of the patient diary and the patient information.

Furthermore, feedback was given in the following meeting of the citizens' forum on how the citizens' comments were incorporated into the further work.

## Trial status
Patient recruitment started in June 2023 and is ongoing. Planned end of patient recruitment is in April 2024.

## ETHICS AND DISSEMINATION
The study will be conducted in accordance to the Declaration of Helsinki in its current version.[38] The University of Würzburg institutional review board approved MicUTI on 16 December 2022 (protocol n. 109/22-sc). The recording, disclosure, storage and analysis of personal data within this clinical trial will take place by the legal provisions of the European Union General Data Protection Regulation (GDPR). To protect the confidentiality,

all data will be collected and stored psudonymised. Pseudonymisation lists will be stored in a secure place at each study sites separately from CRFs.[39] Details of procedures to ensure confidentiality are provided in the online supplemental file.

A trial steering committee is established. The committee is composed by the principal investigator (IG), the study coordinator (PKK) and by internationally renowned experts in the implementation and conduction of pragmatic RCTs in primary care and research methodology (AH, AF, AH, GS, MHE). The committee provides an overall methodological supervision of the study and safeguards the interest of study participants. Any of its members will be informed of amendments to the protocol.

The implementation of MicUTI will generate the needed knowledge to plan a confirmatory, full-scaled trial, intended to reduce the clinical equipoise about rapid POCTs used to inform the clinical management of uUTIs in general practice. Study results will be disseminated through peer-reviewed publications, academic conferences, a report addresses to participating GPs and through a summary report in lay language addressed to patients and the public.

**Author affiliations**
[1]Department of General Practice, University Hospital Würzburg, Würzburg, Germany
[2]Department of Preventive Medicine and Public Health, University of Santiago de Compostela, Santiago de Compostela, Spain
[3]Institute of General Practice and Family Medicine, University Hospital, Ludwig Maximilians University Munich, Munich, Germany
[4]Centre for Academic Primary Care, Bristol Medical School: Population Health Sciences, Department of Community Based Medicine, University of Bristol, Bristol, UK
[5]Institute of General Practice and Health Services Research, School of Medicine, Technical University of Munich, Munich, Germany
[6]Institute of AI and Informatics in Medicine, School of Medicine, Technical University of Munich School of Medicine, Munich, Germany
[7]Institute for Hygiene and Microbiology, University of Würzburg, Würzburg, Germany
[8]Department of Cellular, Computational and Integrative Biology, Interdepartmental Center of Medical Sciences (CISMed), University of Trento, Trento, Italy
[9]Institute of Public Health and Nursing Research (IPP), University of Bremen, Bremen, Germany
[10]Institute of General Practice, Friedrich-Alexander-Universität Erlangen-Nürnberg, University Hospital Erlangen, Erlangen, Germany
[11]Department of Epidemiology and Biostatistics, University of Georgia, Athens, Georgia, USA

**Correction notice** This article has been corrected since it was first published. Author name 'Ildikó Gágyor' has been updated.

**Acknowledgements** We are grateful for all participating general practices. A particular thank goes to Waltraud Flederer, Merle Klanke, Maike Ermster, Kathrin Lasher, Anna-Lena Schnaidt and Petra Hagenbusch for their help in the field implementation. We also thank Alice Serafini, MD, for her genuine advices in the planning of the field implementation. PKK and EB, as PhD and MD candidates respectively, are supported by funds of the Bavarian State Ministry of Science and Arts and the University of Würzburg to the Graduate School of Life Sciences, University of Würzburg.

**Collaborators** Bavarian Practice-Based Research Network (BayFoNet) members include, in alphabetical order: Andrea Baumgärtel, Melanie Bößenecker, Tobias Dreischulte, Stefanie Eck, Ildikó Gágyor, Jochen Gensichen, Alexander Hapfelmeier, Susann Hueber, Merle Klanke, Christian Kretzschmann, Thomas Kühlein, Peter K. Kurotschka, Kathrin Lasher, Klaus Linde, Klara Lorenz-Dant, Marco Roos, Linda Sanftenberg, Antonius Schneider, Stefanie Stark, Til Uebel, Fabian Walter.

**Contributors** PKK and IG conceived the study and are primarily responsible for the study design. GB, together with PKK, planned the randomisation, the statistical analyses and the sample size. PKK wrote the protocol consulting IG, AH, AF, GS, GB, ADH, ASi, ASc and MHE (trial design), OM, T-TL and OK (point-of-care tests and microbiological analyses), IG and CK (data protection and PPI), EB, IE, SS, LW, LS and JG (local implementation, data collection). All authors read and approved the final version of the manuscript.

**Funding** The present study is funded by the German Federal Ministry of Education and Research (BMBF) grant Nr. 01GK1903A 'Development of a sustainable structure for practice-based research networks for the strengthening of general practice'. The funder had no role or authority in the design, or in the writing of the protocol. Moreover, the funder will have no role or authority in data collection, data management, data analysis, interpretation, reporting, writing up or publication of findings.

**Competing interests** None declared.

**Patient and public involvement** Patients and/or the public were involved in the design, or conduct, or reporting, or dissemination plans of this research. Refer to the Methods section for further details.

**Patient consent for publication** Not applicable.

**Provenance and peer review** Not commissioned; externally peer reviewed.

**ORCID iDs**
Peter K Kurotschka http://orcid.org/0000-0003-3750-6147
Gábor Borgulya http://orcid.org/0000-0001-6396-6126
Linda Sanftenberg http://orcid.org/0000-0002-3174-5297
Antonius Schneider http://orcid.org/0000-0002-2847-8626
Lisette Warkentin http://orcid.org/0000-0002-6241-6082
Mark H Ebell http://orcid.org/0000-0003-3228-2877
Ildikó Gágyor http://orcid.org/0000-0002-7974-7603

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
