## [Reviewer comments · BMJ Open]

ARTICLE DETAILS

TITLE (PROVISIONAL)	Dipsticks and point-of-care microscopy to reduce antibiotic use in women with an uncomplicated Urinary Tract Infection (MicUTI): Protocol of a randomised controlled pilot trial in primary care
AUTHORS	Kurotschka, Peter Konstantin; Borgulya, Gábor; Bucher, Eva; Endrich, Isabell; Figueiras, Adolfo; Gensichen, Jochen; Hay, Alastair; Hapfelmeier, Alexander; Kretzschmann, Christian; Kurzai, Oliver; Lam, Thien-Tri; Massidda, Orietta; Sanftenberg, Linda; Schmiemann, Guido; Schneider, Antonius; Simmenroth, Anne; Stark, Stefanie; Warkentin, Lisette; Ebell, Mark; Gagyor, Ildiko

VERSION 1 – REVIEW

REVIEWER	Gibson, W. University of Alberta Faculty of Medicine and Dentistry, Division of Geriatric Medicine
REVIEW RETURNED	17-Oct-2023

GENERAL COMMENTS	Thank you for the opportunity to review this protocol for a pragmatic RCT of POC testing for uncomplicated UTI in women. Overall the trial is justified and well described. However, the age range of participants is wide, from 18 to 70. There will, therefore, be pre, peri, and post-menopausal women within the sample, and the clinical presentation, microbiology, investigation, and treatment of UTI is markedly different between these women. At the very least data on menopausal status of participants should be collected and included in the analysis, or the trial restricted to pre-menopausal women.
--

REVIEWER	Albrecht, Uwe Mediconomics GmbH
REVIEW RETURNED	27-Oct-2023

GENERAL COMMENTS	Recommendation: Minor revision. Manuscript ID: bmjopen-2023-079345 Title: Dipsticks and point-of-care microscopy to reduce antibiotic use in women with an uncomplicated Urinary Tract Infection (MicUTI): Protocol for a randomised controlled pilot trial in primary care The protocol submitted is the planned generation of data in an interesting area of the need for antibiotic treatment in patients with urinary tract infections. The patient population would benefit tremendously if a therapeutic need for standard antibiotic
---

	treatment could be clarified quickly, also with regard to the development of antibiotic resistance. The authors of the paper will obtain an overview of the respective areas in Bavaria and thus obtain a suitable first comparison between the two study arms in established practices, which should usually be the point of contact in this disease field. There are some issues which I like the authors to address: i) Why are so many authors listed here? Study is conducted in Bavaria (Germany). How are the authors from different countries (UK, Italy, US) associated to the study? ii) Page 4 “Methods and analysis / Design”: The study design from the clinicaltrials.gov site describes the study very clearly. Should be adopted here! Experimental: on-site microscopy and test strip-guided management. GPs whose practice is allocated to the intervention will have their management guided by POCTs, namely phase contrast microscopy and urine strips for all patients who agree to participate. No intervention: usual care Practices in the control group will not have their management guided by POCTs. They will provide usual care. Treatment decisions are usually based on symptoms and dip stick test results (i.e. red cells, white cells, nitrites). iii) Page 6 “Methods and analysis / Patient recruitment / enrollment”: How is the procedure for the described “pre-screening” organized? After (pre-)screening, a randomization number is assigned if inclusion in the study is suitable. Screening number specific to practice 01-001 e.g. for the 1st patient in practice 01 and 03-005 for the 5th patient in practice 03 following a randomization number from the list sent to each practice before study start. The procedure should be clarified. iv) Page 9 “Methods and analysis / Control arm-usual care”: Perhaps clarify again here (possibly also above in the study design) what standard care means, i.e. at what values in the dipstick test is antibiotic administration voted for or when is it voted for antibiotic administration without dipstick test. The control arm could be described more clearly in order to emphasize the exact differences between the two therapy arms.  See the point above on study design! v) Page 9 “Methods and analysis / Statistical analyses”: Could be in addition interesting to compare the practices of each arm. vi) Introduction: should be “RCT” instead of “meta-analysis”  “In addition, the authors of the latter RCT focused on prognostic and moderating factors of treatment effects.” vii) Page 13 “Methods and analysis / Sample size considerations”: In this section the abbreviation “unUTI” was used twice, whereas in the manuscript always “uUTI” was used. Should be consistent.
--	--

REVIEWER	Laan, Bart Amsterdam UMC - Locatie AMC, Internal Medicine, Infectious Diseases
REVIEW RETURNED	29-Oct-2023

GENERAL COMMENTS	This is an interesting concept which is evaluate by a pilot trial. The primary outcome is feasibility, which does not have to be evaluated by a cluster RCT. Therefore, the mixed-methods part would be very helpful, but these are reported elsewhere. I have a few minor suggestions. The primary objective is not clearly presented. I suppose this is a pilot study before a 'true' cluster RCT, but what feasibility would be accepted? Or is this pilot
---

	study mainly to have an idea of a valid sample size? Of course, next to feasibility of using the POCT. What will the researcher do with urine that is collected at home and thus analyses could not be carried out immediately after collection? The usual care is only minor reported. Are there guidance / guidelines in Germany that suggest a diagnostic approach for UTIs?
--	--

REVIEWER	Luchristt, Douglas Duke University School of Medicine
REVIEW RETURNED	02-Nov-2023

GENERAL COMMENTS	Abstract: The abstract would benefit from proofreading for consistency. There are inconsistent abbreviations (e.g. POCT vs POC-tests) used in the abstract. Introduction: Appropriate justification for the study was made, however it may be worth considering a more robust example of the current standard of care / usual care to more clearly identify what information (mainly the microscopy) that is being incorporated into clinical decision making and how treatment is applied in each instance. This could also be added to the methods with a more robust description of the usual care arm. Methods: Is there a citation or justification for the determination of a positive on the POC microscopy and dipstick testing? It sounds as if a contaminated specimen could screen positive. Statistical plan is appropriate for the study type.
--

VERSION 1 – AUTHOR RESPONSE

Reviewer: 1

Dr. W. Gibson, University of Alberta Faculty of Medicine and Dentistry

Comments to the Author:

Thank you for the opportunity to review this protocol for a pragmatic RCT of POC testing for uncomplicated UTI in women. Overall the trial is justified and well described.

However, the age range of participants is wide, from 15 to 70. There will, therefore, be pre, peri, and post-menopausal women within the sample, and the clinical presentation, microbiology, investigation, and treatment of UTI is markedly different between these women.

At the very least data on menopausal status of participants should be collected and included in the analysis, or the trial restricted to pre-menopausal women.

Author's reply: We thank the reviewer for this important comment. Primary care relevant guidelines in Germany and Europe do not recommend different UTI management approaches with regard to menopausal status alone if the condition is otherwise uncomplicated.^{1 2 3} In addition, the evidence that justifies this trial was derived from previous primary care trials performed across Europe, which investigated a very similar population, including women aged 16-70.⁴ One trial⁵ even included all women above 18, without any age limit. We adopt similar age cut-offs for

inclusion/exclusion as we believe that this will allow us to be representative of a primary care relevant population.

However, we acknowledge that menopausal status could be relevant with regard to diagnostic accuracy of POCTs, prevalence of bacterial species in urine culture, as well antimicrobial susceptibility of detected pathogens. Following your suggestion, we now plan to collect “menopausal status” as a binary variable during the final electronic medical record review. We plan to consider the menopausal status in the description of the study sample and, if appropriate, in subgroup analyses.

Reviewer: 2

Dr. Uwe Albrecht, Medicconomics GmbH

The protocol submitted is the planned generation of data in an interesting area of the need for antibiotic treatment in patients with urinary tract infections. The patient population would benefit tremendously if a therapeutic need for standard antibiotic treatment could be clarified quickly, also with regard to the development of antibiotic resistance. The authors of the paper will obtain an overview of the respective areas in Bavaria and thus obtain a suitable first comparison between the two study arms in established practices, which should usually be the point of contact in this disease field.

Author’s reply: Thank you for this positive feedback.

There are some issues which I like the authors to address:

i) Why are so many authors listed here? Study is conducted in Bavaria (Germany). How are the authors from different countries (UK, Italy, US) associated to the study?

Author’s reply: Thank you for allowing me to be more specific on the role of the co-authors from other countries. Mark H Ebell, Alaistar D Hay and Adolfo Figueiras (together with Guido Schmiemann, Bremen University, Germany) are members of the trial steering committee. As experts in trial design, complex interventions, and research on common infections in primary care, they gave a substantial contribution to the design of the study, as well as the development of the protocol. Orietta Massidda is a Professor for Microbiology of the University of Trento, Italy. She suggested phase-contrast microscopy as a rapid method to detect bacteria in urine in the first place. In addition, she contributed in the development of study site training, to make phase-contrast microscopy applicable in a general practice setting by previously mostly microscopy-naïve personnel. All authors contributed substantially in the writing of the study protocol and the manuscript.

ii) Page 4 “Methods and analysis / Design”: The study design from the clinicaltrials.gov site describes the study very clearly. Should be adopted here! Experimental: on-site microscopy and test strip-guided management. GPs whose practice is allocated to the intervention will have their management guided by POCTs, namely phase contrast microscopy and urine strips for all patients who agree to participate. No intervention: usual care Practices in the control group will not have their management guided by POCTs. They will provide usual care. Treatment decisions are usually based on symptoms and dip stick test results (i.e. red cells, white cells, nitrites).

Author’s reply: Thank you very much for this comment, we rewrote this section of the manuscript according to your suggestion. Please see the paragraphs “study design” and “Control group-usual care”.

iii) Page 6 “Methods and analysis / Patient recruitment / enrollment”: How is the procedure for the described “pre-screening” organized?

After (pre-)screening, a randomization number is assigned if inclusion in the study is suitable.

Screening number specific to practice 01-001 e.g. for the 1st patient in practice 01 and 03-005 for the

5th patient in practice 03 following a randomization number from the list sent to each practice before study start. The procedure should be clarified.

Author's reply: We have enclosed a detailed explanation of pseudonymization procedures in the supplementary file. The provided supplement also contains more details about the pre-screening list.

iv) Page 9 "Methods and analysis / Control arm-usual care": Perhaps clarify again here (possibly also above in the study design) what standard care means, i.e. at what values in the dipstick test is antibiotic administration voted for or when is it voted for antibiotic administration without dipstick test. The control arm could be described more clearly in order to emphasize the exact differences between the two therapy arms.  See the point above on study design!

Author's reply: Thank you. We clarified what usual care is in Germany in both sections of the manuscript (see answer to your previous comment iii). We now refer to the official clinical guidelines of the German Association of General Practice and Family Medicine.⁶

v) Page 9 "Methods and analysis / Statistical analyses": Could be in addition interesting to compare the practices of each arm.

Author's reply: Thank you for this comment. A detailed statistical analysis plan will be one of the outputs of this project. We will consider this suggestion, as differences may be indeed detected across practices in the same arm.

vi) Introduction: should be "RCT" instead of "meta-analysis"  "In addition, the authors of the latter RCT focused on prognostic and moderating factors of treatment effects."

Author's reply: We added "...of RCTs" to improve clarity. Kaußner et al. 2022 was an individual participant meta-analysis of RCTs, not an RCT.⁴

vii) Page 13 "Methods and analysis / Sample size considerations": In this section the abbreviation "unUTI" was used twice, whereas in the manuscript always "uUTI" was used. Should be consistent.

Author's reply: Thank you very much for this comment, we reviewed the manuscript carefully for inconsistencies and flow.

Reviewer: 3
Mr. Bart Laan, Amsterdam UMC - Locatie AMC

This is an interesting concept which is evaluate by a pilot trial. The primary outcome is feasibility, which does not have to be evaluated by a cluster RCT. Therefore, the mixed-methods part would be very helpful, but these are reported elsewhere.

I have a few minor suggestions. The primary objective is not clearly presented. I suppose this is a pilot study before a 'true' cluster RCT, but what feasibility would be accepted? Or is this pilot study mainly to have an idea of a valid sample size? Of course, next to feasibility of using the POCT.

Author's reply: Thank you for these important comments. We acknowledge that we were not explicit enough in declaring the criteria of feasibility. We now explicitly state that "A confirmatory study without major design changes will be considered feasible if study sites recruit 10 patients each and patients lost to follow up do not exceed 20%" (see "Primary objective").

What will the researcher do with urine that is collected at home and thus analyses could not be carried out immediately after collection?

Author's reply: They will not be considered in this study. POCTs are carried out immediately after specimen collection to avoid growth and false positive microscopy test results. We tried to point this out more explicitly in the paragraph "Intervention arm".

The usual care is only minor reported. Are there guidance / guidelines in Germany that suggest a diagnostic approach for UTIs?

Authors reply: Thank you for this suggestion. We elaborated on usual care throughout the manuscript (please see the paragraphs "study design" and "Control group-usual care") to improve clarity and now refer to the official clinical guidelines of the German Association of General Practice and Family Medicine.⁶

Reviewer: 4

Dr. Douglas Luchristt, Duke University School of Medicine

Comments to the Author:

Abstract:

The abstract would benefit from proofreading for consistency. There are inconsistent abbreviations (e.g. POCT vs POC-tests) used in the abstract.

Author's reply: Thank you for pointing this out. We reviewed the abstract and the manuscript carefully to avoid inconsistencies throughout.

Introduction:

Appropriate justification for the study was made, however it may be worth considering a more robust example of the current standard of care / usual care to more clearly identify what information (mainly the microscopy) that is being incorporated into clinical decision making and how treatment is applied in each instance. This could also be added to the methods with a more robust description of the usual care arm.

Author's reply: Thank you allowing us to clarify this in the manuscript. We elaborated on usual care throughout the manuscript (please see the paragraphs "study design" and "Control group-usual care") to improve clarity and now refer to the official clinical guidelines of the German Association of General Practice and Family Medicine in these section.⁶

Methods:

Is there a citation or justification for the determination of a positive on the POC microscopy and dipstick testing? It sounds as if a contaminated specimen could screen positive.

Author's reply: One of the aims of this study is to explore diagnostic accuracy of phase-contrast microscopy to predict a positive urine culture. Unfortunately, we could not apply previously published methods as they were poorly described to be replicable.⁷ According to the German best practice guidelines for clinical microbiology (Klinische Mikrobiologische Qualitätsstandards)⁸, a higher bacterial concentration in the sample makes a UTI more likely. A UTI is also more likely when only one, or maximum two bacterial species are detected. We adopted this approach and transferred it to phase-contrast microscopy.

For clarity, we added the reference to the German MIQ-2 guidelines in the paragraph "point of care tests".

Statistical plan is appropriate for the study type.

Reviewer: 1

Competing interests of Reviewer: I have no competing interests to declare.

Reviewer: 2

Competing interests of Reviewer: None.

Reviewer: 3

Competing interests of Reviewer: No competing interests

Reviewer: 4

Competing interests of Reviewer: none

References

1. Leitlinienprogramm DGU: Interdisziplinäre S3 Leitlinie: Epidemiologie, Diagnostik, Therapie, Prävention und Management unkomplizierter, bakterieller, ambulant erworbener Harnwegsinfektionen bei erwachsenen Patienten. Langversion 1.1-2, 2017 AWMF Registernummer: 043/044, http://www.awmf.org/uploads/tx_szleitlinien/043-044I_S3_Harnwegsinfektionen.pdf (Accessed: November 27, 2022).
2. National Institute for Health and Care Excellence (NICE), Urinary tract infection (lower): antimicrobial prescribing. NICE Guideline [NG109]. 2018 . [Available: <https://www.nice.org.uk/guidance/ng109/chapter/Recommendations#choice-of-antibiotic>].
3. Urinary tract infection (lower) - women. Clinical Knowledge Summaries (CKS). National Institute of Care Excellence (NICE), 2022 [Available: <https://cks.nice.org.uk/topics/urinary-tract-infection-lower-women/>].
4. Kaußner Y, Röver C, Heinz J, et al. Reducing antibiotic use in uncomplicated urinary tract infections in adult women: a systematic review and individual participant data meta-analysis. *Clin Microbiol Infect* 2022 doi: <https://doi.org/10.1016/j.cmi.2022.06.017>
5. Ferry SA, Holm SE, Stenlund H, et al. The natural course of uncomplicated lower urinary tract infection in women illustrated by a randomized placebo controlled study. *Scandinavian Journal of Infectious Diseases* 2004;36(4):296-301. doi: 10.1080/00365540410019642
6. Deutsche Gesellschaft für Allgemein- und Familienmedizin (DEGAM). S3-Leitlinie: Brennen beim Wasserlassen. AWMF Registernummer: 053-001. 2018 (Available: https://register.awmf.org/assets/guidelines/053-001I_S3_Brennen_beim_Wasserlassen_2018-09-verlaengert_01.pdf, accessed: May 29, 2023).
7. Beyer AK, Currea GCC, Holm A. Validity of microscopy for diagnosing urinary tract infection in general practice – a systematic review. *Scand J Prim Health Care* 2019;37(3):373-79. doi: 10.1080/02813432.2019.1639935
8. Schubert S. MIQ 02: Harnwegsinfektionen: Qualitätsstandards in der mikrobiologisch-infektiologischen Diagnostik: Elsevier Health Sciences 2020.

VERSION 2 – REVIEW

REVIEWER	Albrecht, Uwe Mediconomics GmbH
REVIEW RETURNED	27-Nov-2023

GENERAL COMMENTS	The authors have made positive changes to the manuscript that have increased the quality of this study.
---

REVIEWER	Laan, Bart Amsterdam UMC - Locatie AMC, Internal Medicine, Infectious Diseases
REVIEW RETURNED	30-Nov-2023

GENERAL COMMENTS	Thank you for addressing the comments.
--

REVIEWER	Luchristt, Douglas Duke University School of Medicine
REVIEW RETURNED	12-Dec-2023

GENERAL COMMENTS	Thank you for your edits and clarifications. I look forward to the results of this study.
---